# Wetland fragmentation associated with large populations across Africa

Sani Idris Garba ®[1] ✉, Susanna K. Ebmeier ®[2], Jean-François Bastin ®[3], Danilo Mollicone[4] & Joseph Holden ®[1]

Wetlands provide essential ecosystem services in Africa, yet their extent and fragmentation remain poorly understood. Here we classify African wetlands at 10 m resolution, using seasonal composite imagery and a random forest algorithm. We estimate a total wetland area of 947,750 km² (10% of global wetlands), comprising 46% marshes, 25% swamps, 22% peatlands, 5% seasonal wetlands, and 2% mangroves. Wetland fragmentation is strongly associated with high population densities in countries such as Nigeria, Liberia, Guinea, Egypt, Algeria, and Kenya. African wetlands store an estimated $54 \pm 11$ Gt of carbon, surpassing Europe's 12–31 Gt. If drained, they could release 260 MtC yr$^{-1}$, nearly ten times the carbon sequestration of pristine wetlands (27 MtC yr$^{-1}$). These findings provide a crucial foundation for sustainable wetland management and policy development.

African wetlands are among the most productive ecosystems in the world[1], providing a wide range of services that contribute to human wellbeing, such as the provision of water, food, dry season grazing, and fuel wood. They can support a wide range of flora and fauna and serve as an important carbon pool, sequestrating large amounts of carbon from the atmosphere, thereby regulating climate[2]. Depending on topographic context, wetlands can also play a significant function in flood attenuation and shoreline protection[3–5] and also play a key role in the hydrological cycle[6].

Wetlands are dynamic ecosystems that can be categorized based on their hydrology, soil composition, and vegetation types, each supporting unique ecological functions and biodiversity. Marshes, for instance, are wetlands dominated by herbaceous (non-woody) plants, characterized by periodic or continuous flooding. They can be found in both freshwater and saline environments, where nutrient-rich soils foster diverse flora and fauna. Swamps are wetlands with mineral soils (although some classifications also distinguish organic soil peat swamps), dominated by woody vegetation such as trees and shrubs. These ecosystems experience seasonal or permanent flooding and include coastal mangrove swamps, which are crucial for coastal protection, carbon sequestration, and biodiversity conservation.

Peatlands represent a distinct wetland category, defined by the accumulation of partially decomposed organic matter (peat) due to water saturation. Peatlands are further classified into bogs and fens. Bogs are rain-fed (ombrotrophic) systems that are typically acidic and nutrient-poor, often supporting mosses, shrubs, and sometimes trees. Fens, by contrast, are groundwater-fed (minerotrophic) and more nutrient-rich, allowing for a mix of grasses, sedges, and woody vegetation. Seasonal wetlands, another important type, experience periodic inundation during specific times of the year, followed by dry conditions. These include natural systems like ephemeral ponds and human-made systems such as rice paddies, which harbor species adapted to fluctuating water levels.

Amongst these wetland types, marshes are especially susceptible to anthropogenic pressures because of their accessibility, fertile soils, and proximity to densely populated regions. High demand for agricultural and urban development, combined with inadequate protection measures, makes these wetlands highly vulnerable and often heavily exploited.

In this study, wetlands are broadly classified into five types: swamps (mineral soil dominated), marshes including seasonal marshes are classified under the broader category of marshes, emphasizing vegetation type as the primary distinguishing factor, peatlands

[1]water@leeds, School of Geography, University of Leeds, Leeds LS2 9JT, UK. [2]School of Earth and Environment, University of Leeds, Leeds LS2 9JT, UK. [3]TERRA, Teaching and Research Centre, Gembloux Agro Bio-Tech, Université de Liège, Liège, Belgium. [4]Food and Agriculture Organization of the United Nations, Rome, Italy. ✉e-mail: idrisgarbasani@gmail.com

(encompassing both bogs and fens), seasonal wetlands (including human-made systems and lakes), and mangroves (coastal wetlands dominated by salt-tolerant woody species). This classification framework emphasizes the hydrological, vegetative, and soil characteristics of each wetland type, facilitating effective differentiation and mapping. By recognizing these distinctions, the study captures the ecological and functional diversity of wetlands across Africa, enabling better conservation and management strategies.

Wetlands in Africa are experiencing immense pressure from human activities, the most important being direct drainage and conversion to farmland, diversion of water away from wetlands for agricultural irrigation, population growth and urban expansion into wetland areas, pollution, overgrazing, and hydropower development; there has often been excessive exploitation by local communities[7–9]. A large number of African wetlands are thought to have been heavily modified by overexploitation (e.g., the Yala swamp and Kingwal wetland in Kenya and Nakivubo swamps in Uganda)[2,10] and upstream developments altering the quality and flow of water feeding wetlands (e.g., Hadejia Jam'are floodplain in Nigeria[9]). Many African wetlands have been lost due to agricultural conversion, such as the Ga-mampa swamp in South Africa[11]. However, the current extent of wetland across Africa, at high resolution, is not known, and most continental datasets are very coarse estimates (e.g., 250 m to 1 km resolution)[3,10,12–14]. Small-scale wetlands may have been omitted or overestimated in previous continental mapping studies due to coarse resolution datasets, lack of ground control points and validation[15–19]. It is therefore not known whether the cumulative coverage of small wetlands is important to diverse ecological, climatic, and hydrological functions, and there is a need to ensure appropriate representation of African wetlands for sustainable management and for modeling climate mitigation and biogeochemical cycles. The lack of high-resolution data hinders our estimates of the total amount of carbon stored by these wetlands and estimates of the potential for net carbon uptake or loss from African wetlands at a continental scale. Much wetland carbon is belowground, yet potentially fragile and susceptible to rapid loss with wetland degradation[20]. Wetlands can become divided or separated into smaller, isolated patches or fragments due to both human activities and natural processes, including urbanization, agriculture, infrastructure development, and changes in hydrology. In addition to topographic reasons for the occurrence of small wetland patches in a landscape, formerly larger wetlands can become divided or separated into smaller patches due to natural processes. Wetland fragmentation poses a serious threat to the health and functionality of wetland ecosystems, highlighting the need for conservation efforts focused on preserving and restoring these valuable habitats. Wetland degradation refers to a decline in the ecological integrity and functionality of wetlands, characterized by reduced biodiversity, compromised water quality, and diminished carbon sequestration potential. It occurs due to drivers such as pollution, drainage, excessive water extraction, encroachment by invasive species, and the impacts of climate change. Unlike fragmentation, which involves the physical separation of wetland areas into smaller patches, degradation focuses on the deterioration of the ecosystem's health and its ability to provide essential services, irrespective of spatial continuity. Climate change further exacerbates fragmentation through altered hydrological regimes and extreme weather events.

This study combines spatial mapping, fragmentation analysis, and carbon stock estimation to develop a comprehensive understanding of wetland dynamics across Africa. The primary objectives of this study are to: (1) systematically map the spatial distribution of wetlands across Africa by categorizing them into five distinct types—marsh, mangrove, swamp, peatland, and seasonal wetland—based on ecological and hydrological characteristics; (2) analyze wetland patchiness and assess its relationship with population density, testing the hypothesis that wetlands in highly populated areas are more fragmented, and (3) estimate the total carbon stocks for each wetland type and calculate potential carbon emissions under pristine and drained conditions across various climate zones.

To classify our wetland types, we compile control points for climate zones. The control points are grouped into five wetland types, including marsh (2202), mangrove (1477), swamps (1891), peatland (1580), and seasonal wetland (1054). Here we classify swamps as mineral soil wetlands, while peatlands include fen and bog systems with or without trees (these include what are sometimes referred to as peat swamps). These classes capture critical differences in wetland vegetation, soils and water levels (see Supplementary Tables S2–S6) and importantly are separable using optical and radar-derived indices from freely available satellite datasets[21]. We analyze the relationship between wetland patchiness derived from our map and population data from the Gridded Population of the World database (GPW V4) and test the hypothesis that highly fragmented wetlands are associated with large populations. We use a 10 km grid for fragmentation analysis based on our previous studies that suggested that average continuous wetland patches cover an area of 10–11 km$^2$ [21].

We calculate total carbon stocks for each wetland type by multiplying the total area of the wetland with typical values of carbon stock per hectare estimated by previous studies[22–28]. We then estimate the carbon emissions from different wetland types for two wetland degradation states (pristine and drained conditions) in each climate zone.

## Results

### The current extent of the African wetland

Our high-resolution continental study reveals that wetlands cover ~947,750 km$^2$ of Africa (excluding deep water bodies), which constitutes ~3% of the total land area. Marshes, covering 436,743 km$^2$ (46% of total wetlands), are more extensive than swamps, which account for 231,776 km$^2$ (24%). Peatlands cover 208,842 km$^2$ (22%), while seasonal wetlands (5%) and mangroves (3%) have the least coverage. Most of these wetlands are concentrated in western and central parts of Africa (Fig. 1a), where there is a high amount of rainfall throughout the year. However, some important wetland complexes (regions where there are two or more wetland types clustered together) are situated in North Africa, such as in the Nile region. The largest wetland complex is located in the Congo region of central Africa covering about 278,450 km$^2$, which contains the most extensive peatland area (165,250 km$^2$) in the entire continent (Fig. 1e). Other important wetland complexes are situated in southern Sudan (the Sudd) (67,150 km$^2$) (Fig. 1d), the Zambia (43,170 km$^2$), Angola (46,072 km$^2$) and Nigeria (47,130 km$^2$).

### Distribution of wetland across African climate zones

We present the spatial distribution of different wetland types based on high resolution image processing, according to the five main climatic regions in Africa based on the Köppen–Geiger classification[29]: Tropical wet (TW), tropical wet and dry (TWD), semi-arid (SARD), arid or desert (ARD) and mediterranean subtropical climate (MED). The overall accuracy of the trained algorithm compared to validation ground control data is higher for wetlands in the TWD region (89%), with mangrove and marsh well distinguished from other wetland classes with producer's accuracies exceeding 80% (Supplementary Table S2). TW has the most extensive wetland, hosting 57% (~448,210 km$^2$) (Fig. 2) of the total wetland area in Africa. Peatlands (37%) and swamps (34%) are the most dominant wetland types of TW, which cover 165,950 km$^2$ and 153,580 km$^2$, respectively. Mangroves (2%) and seasonal wetlands (0.5%) are the least common wetland types in TW, covering only about 14,000 km$^2$. The largest climate region is the TWD, covering up to 38% of the total area of Africa. Wetlands in this region constitute only 3.2% (~362,980 km$^2$) (Supplementary Table S1) of the total area with 52% being marshes (Fig. 2a). This region has a distinct climatic feature with alternating wet and dry periods throughout the year, which plays a significant role in the formation of different wetland conditions and

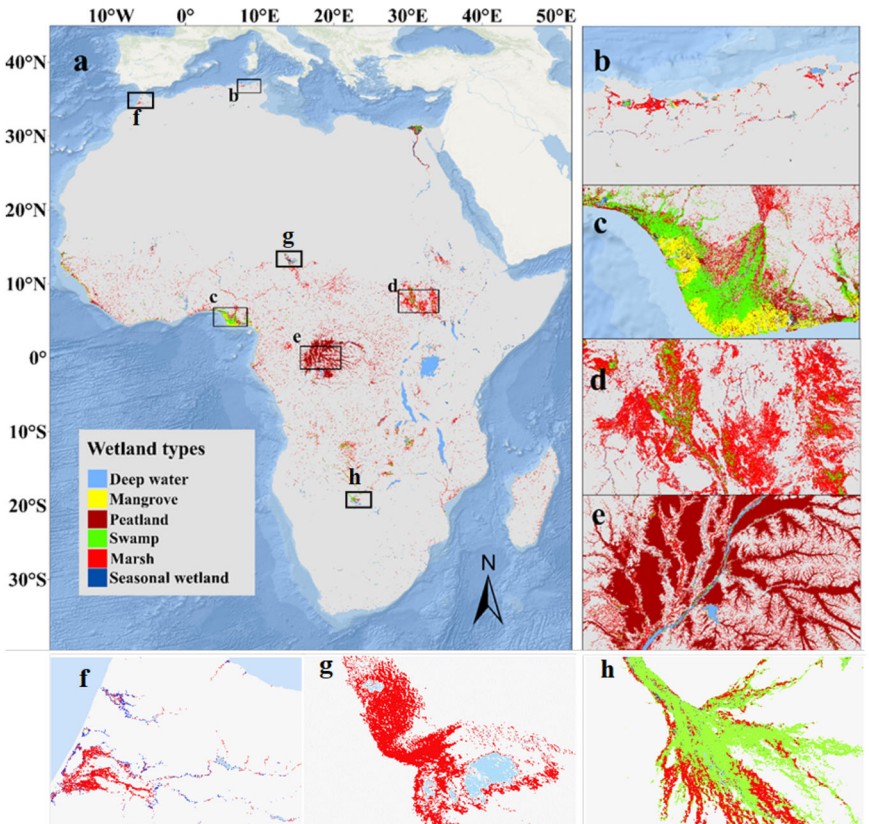

**Fig. 1 | The distribution of wetland in Africa, at 10 m resolution.** Continental distribution derived from classification of a combination of Sentinel-1 and Sentinel-2 composites between January 2020 and January 2021, The Basemap data used provided by Esri (**a**) and extensive wetland complexes in **b** northern Algeria and Tunisia, **c** Nigeria, **d** South Sudan, **e** part of the Congo basin, **f** Morocco, **g** Chad, **h** Botswana.

variability across the season. Thus, TWD has a higher amount of seasonal wetland cover relative to other climate zones (Fig. 2b). SARD is characterized by 250–500 mm of rainfall throughout the year, covering 6,700,000 km² (22%) of Africa. Only about 1.4% of SARD is covered by wetland, of which seasonal wetlands are the dominant type. ARD is the second largest climatic region in Africa, extending up to about 9,000,000 km², and has the lowest wetland coverage (0.4%).

## Wetland fragmentation and human population

We develop a wetland fragmentation and population index (WFPI) by overlaying the gridded population layer with the gridded wetland fragmentation layer using the fuzzy overlay method (see "Method"). The fragmentation index is an indicator of regions with a high number of wetland patches (based on the 10 m resolution data) per 10 km grid, and the population index is a count of persons per kilometer grid downscaled into a 10 km grid, indicating areas of high concentrations of population. Our WFPI shows areas where fragmentation is coincident with humans (Fig. 3) using 10 km grid cells across Africa.

We identify nine regions with a WFPI value indicating highly fragmented grid cells (80–226 wetland fragments per 10 km²) related to large population size (40,000–300,000 persons per 10 km²). Six of these regions are in West Africa (Nigeria, Liberia, Gabon, Guinea, and Cameroon), two in North Africa (Egypt and Algeria), and one in East Africa (Kenya). In West Africa, areas such as Rivers State and Lagos in Nigeria, and Monrovia in Liberia, have the highest WFPIs of 0.89, 0.76, and 0.83, respectively (Table 1). These areas are characterized by high population growth associated with urban expansion, thereby increasing pressure on nearby wetlands, mainly coastal mangroves and swamps. Other areas with high WFPI include Conakry in Guinea (0.68), Alexandria in Egypt (0.66), Algiers in Algeria (0.61), and Murang'a in Kenya (0.59) often associated with agriculture encroaching on

wetlands in these regions[30–32]. Our index indicates that a total of 13,021 km² of wetlands may be heavily threatened by human activity within Africa (WFPI > 0.5), and about 28,724 km² of wetland occurs in populated areas, which suggests a moderate risk of human interactions (WFPI 0.3–0.5). However, large wetland areas with a high concentration of fragments (for example the Congo basin wetlands) that are far away from settlements or sparsely populated show little or no relation between fragmentation and human populations (Fig. 3). The high concentration of fragments in the Congo basin are thought to be geomorphologically and climatologically controlled rather than driven by human activities[25], though these peatlands could be highly sensitive to human-induced fragmentation in the future. The low fragmentation in the Great African Lakes region is likely due to the hydrological stability and the presence of large, continuous wetland systems. These wetlands are naturally resistant to fragmentation, while human activities like agriculture and urbanization are primarily confined to upland areas.

## Carbon stock in African wetlands

Healthy wetlands can store large amounts of carbon, but the quantity of carbon stored varies among different wetland types[20,22]. Among these wetland types, peatlands are thought to have the highest carbon stock followed by mangroves, swamps and marshes[22–24,33]. We use the IPCC mean carbon stock for each wetland type, according to different climate zones, to estimate the total carbon stored in the four wetland types of Africa. Our continental map indicates that African wetlands contain 54 ± 11 Gt of carbon, which is around 5% to 9% of wetland soil carbon stored globally (520–710 Gt C)[20], and higher than that of European wetlands (12–31 Gt)[34] based on differences in wetland area. Peatlands store about 41% of this African wetland carbon, while 28% is stored in marshes, 27% in swamps, and 3% in mangroves (Fig. 2d).

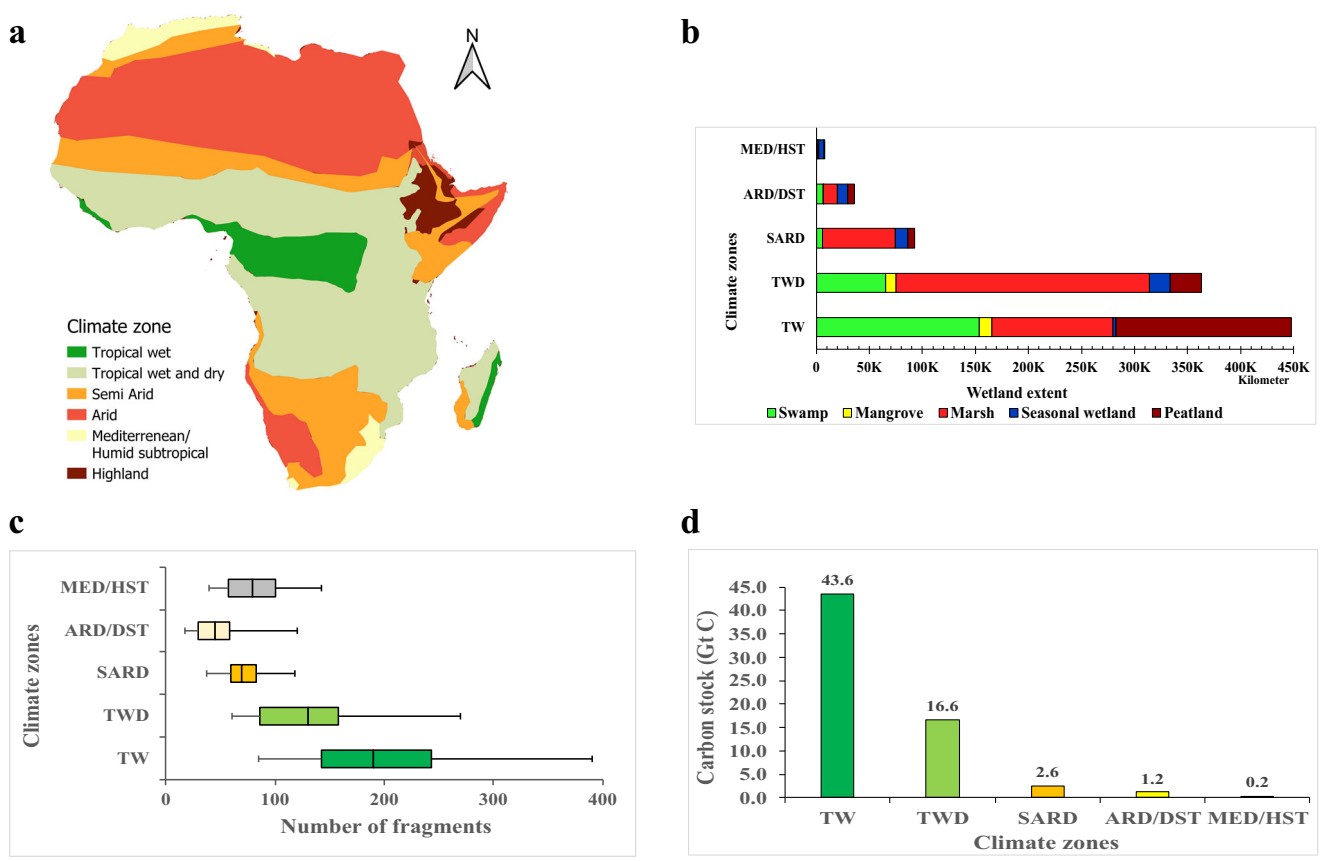

**Fig. 2 | Distribution of wetland types in different climatic zones. a** Division of Africa into different climate regions, **b** estimate of the areal extent of wetland types in each climate zone, **c** the intensity of wetland fragmentation in each zone per 10 km grid, **d** carbon stock in wetlands for each climate zone.

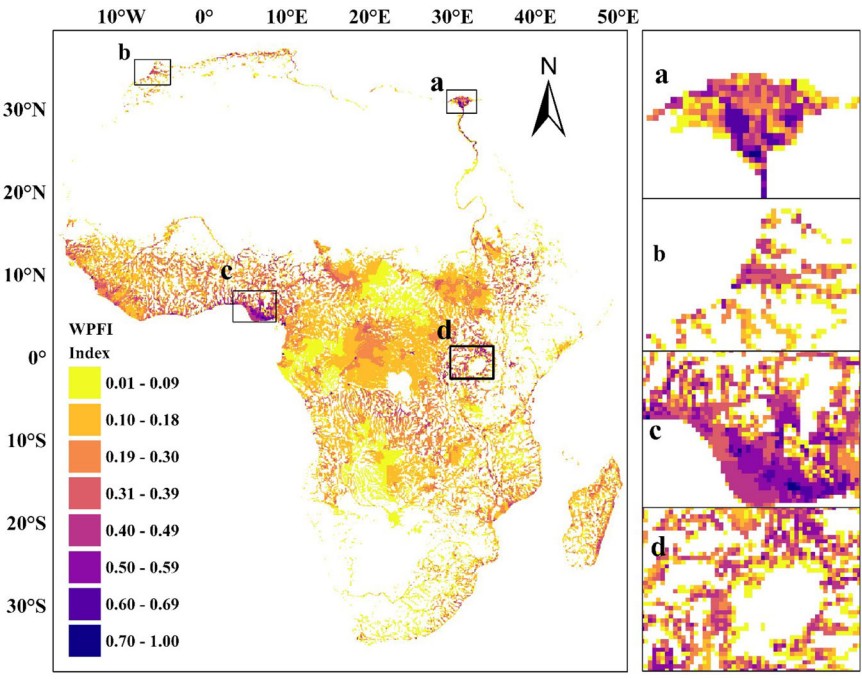

**Fig. 3 | Gridded wetland fragmentation and population index for 10 km cells across Africa showing areas where fragmentation is associated with population (values closer to 1).** The insert map **a–d** shows areas of high WFPI value. Data for wetland cover uses our 10 m dataset to determine fragmentation within each 10 km cell. Data for population density uses 1 km data reprofiled to 10 km.

**Table 1 | The location, number of fragments and population of high WFPI 10 km² grid cells across Africa for the year 2020**

| Location | Country | Population per grid cell | Fragmentation per grid cell | WFPI |
|---|---|---|---|---|
| Rivers state | Nigeria | 130,698 | 209 | 0.89 |
| Lagos | Nigeria | 303,143 | 107 | 0.76 |
| Greater Monrovia | Liberia | 136,475 | 185 | 0.83 |
| Alexandra | Egypt | 168,943 | 94 | 0.66 |
| Algiers | Algeria | 220,546 | 76 | 0.61 |
| Muranga | Kenya | 72,349 | 61 | 0.59 |
| Conakry | Guinea | 98,844 | 174 | 0.68 |
| Littoral | Cameroon | 67,846 | 173 | 0.57 |
| Estuarie | Gabon | 41,947 | 226 | 0.55 |

### Net carbon uptake or loss from African wetlands

We estimate the contribution of African wetlands to the global carbon budget across each climatic region using empirically derived emission rates for selected wetland types for which data are available (see Method). We use two approaches, First, we use the default emission factor from the IPCC emission factor database to calculate total carbon emissions from wetlands under two conditions (drained or natural). Using our high resolution map of wetlands in Africa, we calculate that drained peatland, mangrove and marsh are capable of emitting 260 Mt C yr$^{-1}$ (936 Mt $CO_2$ yr$^{-1}$ equivalents) which is equivalent to 2.4% of global net annual $CO_2$ emissions[35] and almost ten times the mean net annual uptake under natural conditions of 27 Mt C yr$^{-1}$ (98 Mt $CO_2$ equivalents yr$^{-1}$) by these wetlands. Wetlands within high WFPI areas, under drained conditions, could release 10.3 Mt C yr$^{-1}$ (37 Mt $CO_2$ equivalents yr$^{-1}$).

The net wetland carbon flux varies according to water level[36,37]. Therefore, in our second approach, we use the emission factor for different wetland types at various water levels obtained from Zou et al.[37] to estimate the carbon flux for peatlands, marshes, and mineral soil swamps. The six categories of water level range from −3 to 2 (WTL-3 ≤ −70 cm; −70 cm < WTL-2 ≤ −50 cm; −50 cm < WTL-1 ≤ −30 cm; −30 cm < WTL0 ≤ −5 cm; −5 cm < WTL1 ≤ 40 cm; and 40 cm < WTL2), where negative values indicate depth below the surface, while positive values indicate ponding. At water level −3 we estimated that African wetlands will have a net release of 310 Mt $CO_2$ equivalent yr$^{-1}$, while for water level −2 they will emit 115 Mt $CO_2$ yr$^{-1}$ and 46 Mt $CO_2$ yr$^{-1}$ for water level −1, while 91 Mt $CO_2$ yr$^{-1}$ will be taken up by African wetlands when the water level is at level 1.

## Discussion

Our estimate of wetlands in Africa (947,750 km²) is larger than that of the coarser global wetland dataset by CIFOR (859,278 km²) and that of GLWD (934,481 km²). The variation may be due to the coarser resolution imagery used to produce previous global wetland maps, which may result in misclassification and omission of small-scale wetlands. This inconsistency highlights the importance of using high-resolution data to improve the estimation of wetlands, which in turn can be used to develop policy and monitoring to protect wetlands. Our study shows close similarities with smaller geographical scale studies, such as the peatland map of Angolan highlands constructed by Lourenco et al.[38] (see Supplementary Fig. S6).

There was high confusion in discriminating between mineral soil swamps and peatlands, especially in the TW region, with users' and producers' accuracy below 70% and 80%, respectively. There is also a common confusion amongst other wetland classes, such as swamps and mangrove, marsh, and seasonal wetlands, due to similarities in their visual and spectral signatures. The low accuracy in the Arid region (Supplementary Table S5) was a result of confusion in discriminating

swamp and peatlands along the Nile area, due to the presence of a peat deposit within the swamps. Similar confusion occurs in TW and TWD due to peat deposits in swamps, so our method did not perform well in discriminating between non-peatland swamps and peatland.

The WFPI analysis generally identifies sites where the impact of human activities results in patchiness of the surrounding wetlands. The overlap between fragmentation and high population density indicates an increased susceptibility to future human impacts. It should be noted that some extensive wetland regions with a high density of fragments, such as those in the Congo Basin, show minimal correlation between fragmentation and human populations, especially in areas distant from settlements or with sparse human activity. The fragmentation in these wetlands is largely attributed to geomorphological patterns and climatological processes rather than anthropogenic influences. However, these peatlands remain susceptible to potential future fragmentation driven by human activities, emphasizing the need for future studies that examine wetland change and detect changes in fragmentation to support vigilant conservation and management strategies. Our map can be used as a baseline to monitor and assess wetland changes over time at a fine scale (10 m resolution). It should also be noted that our method can now be used to generate timeseries observations for analyzing human-driven and natural wetland changes, as well as their fragmentation, supporting future remotely sensed observations on the success of different sustainable wetland protection policies. We use detailed ground control points specifying wetland types and locations, ensuring that incorporating topographic indices would not substantially affect classification accuracy. However, we aim to refine the models further in future research by integrating topographic variables to enhance their precision and applicability. Our future work will be concentrated on gathering more and better-quality ground control data to support some future time series analyses.

We explore the possible impact of African wetlands on global climate through net carbon uptake/loss under natural and a range of drained conditions. We find that the three selected wetland types (peatland, mangrove, and marsh) under drained conditions could contribute up to 3% of global net annual carbon loss, a value which might be much higher if data for emissions from other wetlands become available and included in the estimation. Human activities have been widely reported to be a key driver of wetland degradation[39–41]. The degradation of wetland is often related to deeper water tables, which leads to increased decomposition and release of carbon to the atmosphere[42,43]. We find that wetlands, which are currently highly fragmented in heavily populated areas of Africa, have the potential to release $CO_2$ equivalent to 0.6 % of total global annual emissions. Hence, protection of African wetlands, particularly in tropical wet (TW), tropical wet and dry (TWD) regions, and most areas with high WFPI where the largest carbon stocks and greatest net C emission potential are to be found, will be important for managing future land-based emissions. Although land use was indirectly considered in the carbon loss estimations, a comprehensive evaluation of its specific impacts was outside the scope of this study. Future investigations should integrate detailed land use data to better understand and quantify its contribution to wetland fragmentation and degradation.

Our analysis of African wetlands provides a high-resolution insight as to their extent, condition and their potential contribution to the global carbon balance, providing data critical for both improving land-surface climate models and for sustainable wetland conservation.

## Methods
### Datasets

**Ground control points.** We collated data on the location and characteristics of wetlands across Africa from reliable sources, including the Food and Agriculture Organization (FAO) global dryland

assessment database, Global Peatland Database (GPD), journal papers, academic reports, and NGO reports. We verified each data point and screened them to exclude any coordinates that were inaccurate, mislabeled, or inconsistent by using visual interpretation of very high spatial resolution digital globe images (>1 m pixel) made available for visualization through Google Earth. Our final dataset used 8204 control points for different wetland types in Africa. These control points were sorted based on the climate zones in Africa and assigned to either training or validation points (Supplementary Fig. S4). Thus, the control points were grouped into an equal number of training and testing points to ensure robust accuracy assessment[16,44].

**Satellite data.** Sentinel-1 and -2 satellite images covering the entire study area for the period of January 2021 to December 2021 were available through the Google Earth Engine platform (GEE) at 10 m resolution. We used the Ground Range Detected interferometric wide-swath with a pixel spacing of 10 m Sentinel-1 images acquired in dual-polarization (VV/VH) and pre-processed as a Level-1 data product, with an average acquisition interval of 12 days. A total of 5728 Sentinel-1 images in ascending order were collected for the study area. Sentinel-2A and 2B Top of Atmosphere reflectance data with 13 spectral bands were obtained through the GEE. We used blue (0.496 μm, band 2), green (0.560 μm, band 3), red (0.665 μm, band 4), near infrared (NIR, 0.835 μm, band 8), and short-wave infrared 2 (SWIR2 2.202 μm, band 12) bands. Sentinel-2 images with cloud cover of <20% were selected from January 2020 to January 2021 which resulted in a total of 13596 images[45–49].

**Population data.** We obtained information about population distribution from the Gridded Population of the World database (GPW V4) provided by Center for International Earth Science Information Network (https://sedac.ciesin.columbia.edu). The GPW dataset has an approximate resolution of 30 arcsec, equivalent to 1 km at the Equator, that contains global population counts, density, urban/rural status, age and gender structures with more than 12,500,000 input units maintained by NASA's Socio-Economic Data and Applications Center (SEDAC). The population input data were collected at the finest resolution available from the '2010' round of censuses, which occurred between 2005 and 2014. The data were used to produce population estimates for the years 2000, 2005, 2010, 2015, and 2020 (https://earthdata.nasa.gov/data/catalog?keyword=gpw-v4/methods). We selected the population estimates for 2020 for our analysis.

**Mapping of wetland extent**
To accurately delineate the wetlands of Africa we classified the continent into different major zones according to the climatic and ecological features. These zones include tropical wet (TW), tropical wet and dry (TWD), mediterranean subtropical (MED), semiarid (SARD), and desert or arid (ARD). We grouped the control points for each wetland type based on these climate zones. TWD consists of 3120 control points, TW includes 2550, SARD has 1144, ARD contains 848, and MED comprises 536. We processed the images collected from Sentinel-1 and -2 images for the period of 2020–2021 to develop optical and radar indices for each climate zone. The optical variables used include spectral bands 2 (blue), 3 (green), 4 (red), 8 (NIR), 11 and 12 (SWIR), the normalized difference vegetation index (NDVI), normalized difference water index (NDWI), modified normalized differential water Indices (MNDWI), and tasseled cap wetness index (TCWI). SAR variables included vertically transmitted, vertically received SAR backscattering coefficient (σ⁰VV), vertically transmitted, horizontally received SAR backscattering coefficient (σ⁰VH), and the normalized difference ($Ndiff \frac{VH-VV}{VH+VV}$) and ratio indices ($Nratio = \frac{VV}{VH}$) for the wet and dry season. We then undertook a variable importance analysis[50–57] for each climate zone to select the most important variables to input into the final classification. The highest importance is typically placed on the variables that contribute the most to reducing the model's

impurity or error. We identified these variables by computing variable importance in an RF algorithm using the mean decrease impurity (MDI), which measures how much each variable improves the model's predictive performance. We ran the variable importance algorithm five times before finally selecting the variables with higher reduction in impurity considered as higher importance. For all images in the arid and semi-arid region, we extracted the maximum pixel values, while a median value was used for other regions to enhance identification[58]. Finally, we applied a Random Forest (RF)[59,60] algorithm to classify and validate wetland types in each climate zone.

To produce the forest tree in RF we needed to identify the two important parameters: the number of decision trees to be generated (Ntree); and the number of variables to be selected and tested for the best split when growing the trees (Mtry). The parameter Ntree was assessed for the values of 100 – 600: a value of 500 was selected as error rates for all classification models were constant beyond this point. In this study, we used the combined SAR and Optical indices as input variables.

**Classification map accuracy and uncertainties**
We undertook the classification of wetlands according to each climate zone using the RF classifier. The control points for different wetland types were compiled for each climate region separately to classify the input variables developed for each region. To accurately classify the wetland types based on their distinctive features in a particular region, the input variables were extracted from composite images constructed from different pixel values over a particular period of the year. In ARD, seasonality is often a key property of wetlands. We therefore used the variables extracted from seasonal composites of maximum pixel value to train the RF classifier (Table 2). Seasonal wetlands were also identified using the maximum value from our seasonal composites of wet and dry seasons in each climate zone. For the TWD and MED zones, the variables constructed for the wet and dry season from the mean pixel value composites were used to train the RF classifier (Table 2). The accuracy of RF classifications for each climate zone was assessed using cross-validation by splitting the control points into two halves (50% training and 50% testing points) (Supplementary Fig. S4), spatially selected for each climate zone from each class on a random basis. Our accuracy estimation matrix included the overall accuracy (OA), Kappa coefficient, producer accuracy, and user accuracy. We selected the Kappa coefficient in this analysis due to its widespread application and interpretability in evaluating agreement between observed and predicted classes, making it a reliable metric for wetland classification studies. Overall accuracy determines how well the classification algorithm performed, which can be measured by dividing the total number of correctly identified sample point by the total number of the testing points (Supplementary Tables S2–S6). We evaluated our uncertainties by comparing classifications made using the entire control point dataset with those produced using only a subset of control points selected at random for each wetland class in each climate zone. The uncertainties were associated with our classification accuracy, high confusion between wetland classes (e.g., swamps and peatlands), and limitations in the number of control points. A common issue with the gridded population data is a misallocation of population to areas outside urban areas. These errors were minimized by down-sampling the 1 km gridded population and taking the average population within 10 km grids.

**Wetland fragmentation and population index (WFPI)**
We compared the distribution of wetland fragments and population at the same cell size across 10 km × 10 km grid areas. For our analysis, we used only the count of wetland fragments estimated at a resolution of 10 km to allow comparison to the gridded population data at 10 km resolution. This resolution was selected because it was found to be the mean dimension of wetland fragments from an earlier study in southern Nigeria[21].

**Table 2 | Composites and input variables for each region**

| Study region | Composites | | B8 | B12 | NDVI | NDWI | MNDWI | TCWI | VV | VH | $\dfrac{VV}{VH}$ | $\dfrac{VH-VV}{VH+VV}$ |
|---|---|---|---|---|---|---|---|---|---|---|---|---|
| | Date | value | | | | | | | | | | |
| Tropical wet | Jan - Dec 2021 | Mean | | | | | ■ | ■ | ■ | ■ | ■ | ■ |
| Tropical wet and dry | Jan - Dec 2021 | Mean | | | ■ | ■ | ■ | ■ | ■ | ■ | ■ | ■ |
| Mediterranean/Humid subtropical | Jan - Dec 2021 | Mean | | | ■ | | ■ | ■ | ■ | ■ | ■ | ■ |
| Semi-arid | Jan - April 2021 (wet) | Maximum | ■ | | ■ | ■ | | ■ | ■ | | ■ | ■ |
| Arid/desert | Jan - April 2021 (wet) | Maximum | | ■ | | ■ | ■ | ■ | ■ | ■ | | ■ |

The shaded box indicates the selected variables of highest importance used as input for final classification in each region

We sought to identify the association of wetland fragmentation with human population. We use a fuzzy logic approach to create a membership rank for the fragmentation grid and population grid (ranging from 0 to 1), with 0 representing the lowest membership and 1 the highest membership in increasing order. Lower membership indicated grid cells with less fragments or which are sparsely populated, while grid cells with a large number of wetland fragments or which are densely populated were assigned to a higher membership group. Finally, we overlaid the gridded fragmentation membership layer with the gridded population membership layer to quantify the coincidence of wetland fragments and human population. Higher WFPI indicated interaction of dense population with wetlands, resulting in patchiness within the grid cells.

**Population grid.** The population grid was created by transforming 1 km resolution population data obtained from the GPW V4 data using 10 km grid reference cells across the continent of Africa. We classified the cells in different class ranges from lowest to highest based on the population count in each grid cell (Supplementary Fig. S1). Most of the grids with dense population were located near major city centers or close to river networks. We used this grid as an input for the fuzzy membership transformation.

**Fragmentation grid.** To create the fragmentation grid, we converted the classified wetland raster to polygons using the conversion tool in ArcGIS Pro. We used an algorithm similar to spatial aggregation by overlaying a 10 km × 10 km grid on the original 10 m resolution fragmentation map. For every 10 m cell within a 10 km grid, the number of unique wetland fragments was calculated. We then identified and labeled distinct fragments within each grid. The fragment count within each 10 km grid cell was computed to derive a metric of the total number of fragments. The total number of fragments per grid cell was used to group the cells into eight groups from low to highly fragmented (Supplementary Fig. S2). The total fragment in each cell was calculated by:

$$Frag_{grid} = \sum_{i}^{n} Grid_i \qquad (1)$$

where Frag$_{grid}$ is the fragmentation grid (10 km), n is the number of fragments in grid cell $i$, and $i$ is the code of the grid cell.

**Fuzzy membership.** We transformed the population and fragmentation grid into a fuzzy membership layer scaled from 0 to 1. 0 indicated grid cells that are not members of any set while 1 was assigned to grid cells with full membership. We used the fuzzy linear membership function to transform the input values linearly on the 0 to 1 scale, with 0 being assigned to the lowest input value and 1 to the largest input value (Supplementary Figs. S3 and S4). All of the values in between receive some membership value based on a linear scale, with the larger input values.

**Fuzzy overlay.** The Fuzzy Overlay tool is used to analyze the possibility of a phenomenon belonging to multiple sets in a multicriteria overlay. It determines whether a phenomenon is a possible member of a particular set and analyzes the relationships between the membership of multiple sets. We used the "fuzzy And" function to find the relationship between the population and fragmentation membership layer. We overlaid the population grid with the fragmentation grid using a fuzzy overlay tool[61]. This allowed us to analyze the relationship between the multiple members set from each grid layer. Stronger relationships are found between higher membership sets while lower membership sets show weak relationships. Coincidence of a dense population grid (higher population membership grid) with a highly fragmented grid (higher fragmentation membership grid) resulted in the highest WFPI region.

**Carbon loss estimation**

We used the $CO_2$ emission factor provided by the IPCC Wetland Supplement guidance[28] to estimate the amount of carbon loss from each wetland type for different climate zones in Africa. An emission/removal factor is a coefficient that quantifies the emissions or removals of a gas per unit area. It is calculated based on a sample of measurement data, averaged to develop a representative rate of emission for a given activity level under a given set of operating conditions. We multiplied the total area of each wetland type with its corresponding emission/removal factor across the different climate zones for two assumptions: (1) the wetlands are in pristine condition; (2) wetlands are drained. The $CO_2$ equivalent emission was calculated by:

$$CO_{2\ emission\ i} = \sum_{i,c,y}(EF*Area) \qquad (2)$$

where $CO_2$ equivalent emission is the annual net carbon emission/uptake from a wetland type in tonnes $CO_2$ $yr^{-1}$, area is the land area of drained organic soils in a land-use category in climate domain c, in ha and EF is the emission factor for drained organic soils, by climate domain c, in tonnes C $ha^{-1}$ $yr^{-1}$.

We also adopted the empirical function of Zou et al.[37] to estimate the wetland carbon flux using water level as a function of carbon emission. The equation below was used to calculate the carbon flux from selected wetland types under different moisture regimes in different climate zones:

$$Emission_y = \sum \left( Efs_{ijk} \times Area_{ijy} \right) \qquad (3)$$

where $Efs$ is the emission factors, i is the climate zone, $j$ is the water-table level (coded −3 to 2), $y$ is the year, and $k$ is the wetland type.

## Carbon stock estimation

The total amount of carbon stored by each wetland type in Africa was evaluated using the acquired data of carbon stock per hectare from the studies by refs. 22–24,26–28. To calculate the total amount of carbon stored by each wetland, we multiplied the total area of the wetland by the value of carbon stock per hectare: Wetland carbon = total wetland area (hectare) * carbon stock (t C $ha^{-1}$).

## Data availability

Gridded Population of the World (GPW V4) data (Center for International Earth Science Information Network) are available at https://sedac.ciesin.columbia.edu. The Carbon Emission factor data is available at https://www.ipccnggip.iges.or.jp/EFDB/find_ef.php?ipcc_code=3.B.4&ipcc_level=2. https://doi.org/10.1038/s41561-022-00989-0. The main data supporting the findings of this study are available as Supplementary Information files. The shapefiles for the 10 m resolution wetland map of Africa are available from the University of Leeds data repository (DOI to be added upon paper acceptance).

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

## Acknowledgements

This project was funded by a studentship awarded to SIG (PTDF/ED/PHD/1301/17) under the Petroleum Technology and Development Fund (PTDF), Nigeria. The photo-interpretation dataset was part of the global dryland assessment, which was conducted in the region by the Food and Agriculture Organization and the National Space Research and Development Agency of Nigeria. Ground control data points for some peatlands in East Africa were extracted from the Global Peatland Database (GPD) collated by Greifswald Mire Center and we specifically acknowledge support from Alexandra Barthelmes. S.K.E. is supported by a NERC Independent Research Fellowship (NE/R015546/1) and is a member of the NERC-BGS Center for the Observation and Modeling of Earthquakes, Volcanoes, and Tectonics (COMET).

## Author contributions

The paper was conceived by J.H., S.K.E., and S.I.G. The analysis and manuscript drafting were designed and conducted by S.I.G., with editing provided by the supervisors J.H. and S.K.E. J.F.B. and D.M. provided data used for ground truthing.

## Competing interests

The authors declare no competing interests.
