## [Peer Review file · Nature Communications]

Wetland fragmentation associated with large populations across Africa

Corresponding Author: Mr Sani Garba

Version 0:

Reviewer comments:

Reviewer #1

(Remarks to the Author)

Thank you for considering me as an reviewer for this study on the large-scale fragmentation of wetlands across the African continent. The work presents significant content on the theme of wetland protection. Overall, this study provides important results, with well-written paragraphs and a logical flow. However, in my humble opinion, the new information provided still requires significant changes and additions. This raises questions about its suitability for publication in a journal like Nature Communications. Therefore, I propose major revisions to the article, with the possibility of submission to another Nature journal, such as Scientific Reports, for example.

You will find my comment in the attached file.

Kind regards

Reviewer #2

(Remarks to the Author)

Overview:

This paper addresses the lack of high-resolution data on Africa's wetlands - the extent, the intactness of the wetlands and carbon storage in the wetlands. There is an abundance of information in this paper.

Using this wetland map, and the GPW V4 data used to map of human densities, the authors have derived a measure of wetland intactness/fragmentation and the volumes of carbon stored in the wetlands of Africa.

In our opinion, the highlight of this paper is the production of a 10m resolution map of Africa's wetlands. This is a valuable output which is likely to improve on previous maps with lower resolutions that cannot pick up all the smaller wetlands.

General comments/questions:

- o The paper does not have a specific question to be answered or hypothesis to be tested.
- o It is a multi-disciplinary study – involving computer techniques, cartography, quantification of carbon etc. However, the disciplines do not link seamlessly. It seems as if there are different degrees of rigor in assessing accuracy and precision at different levels in this study. For instance, the wetlands are mapped at a 10 m resolution, but the human distribution patterns at a 10 km resolution. The issues in bringing together these resolutions are not adequately dealt with. There needs to be some understanding of where humans settle. Empirically their distribution is a narrow linear pattern along rivers and in floodplains which is difficult to evaluate when using a 10 km grid for human distribution. It would be good to give some analysis of this.
- o We also have a concern that there may be a compounding of errors – from population censuses, to allocation of densities in 10 km grids, and errors relating to overlapping these on wetlands mapped at a small scale. This is something that does need to be addressed.
- o Overall, the study would benefit by having a discussion on the processes involved – in wetland fragmentation and human settlement patterns. And then, what are the global impacts of such fragmentation.

Mapping of wetlands

o With the mapping of wetlands, we question the classification of the wetlands into the categories the authors have used. There needs to be a clear definition of each wetland type. What is the difference between swamp and marsh? Swamp is usually used to describe a wetland with trees while marsh is shorter vegetation. Papyrus wetlands are a feature of large areas in Africa – are they swamp or marsh? And what distinguishes wooded wetland from seasonal wetland? And peatlands from marsh?

How is a 'seasonal wetland' determined? The term 'wetland complex' is used – what is meant by this?

o Verification: How does the wetland extent compare to other wetland maps generated for specific regions in Africa with smaller geographical extents e.g. Lourenco et al. 2022 mapped the Angolan highlands peatlands. How does this (or other similar maps) compare with the mapping produced for this study? For this study the verification is at a density of one point per 200 km². Is this adequate for such a fine-scale resolution map covering such variable conditions as the whole of Africa?

Mapping of people

o People do live where there is water and wetlands. This is especially the case in regions where water is not abundant. So, one would expect a correlation between human numbers and wetlands. Has there been any change in this over time? Or is it just that, as human numbers increase, so this effect is more pronounced. This can only be shown by having a time series – to detect if this human distribution pattern stays the same or changes over time, and is there a corresponding change in wetland fragmentation?

Fragmentation

o There are causes of wetland fragmentation that do not relate to human population density that need to be discussed? These include the downstream impacts of dams, the impacts of lowering of groundwater table, the impacts of mining and large-scale agriculture.

o To separate natural fragmentation patterns (e.g. due to the nature of topography) from human-caused changes it is necessary to show the relationship over time between numbers of people and a change in fragmentation/transformation. Sentinel 2 was fully commissioned by 2017 – is it within the scope of this study to compare wetland extent between 2017 and 2020? Or at a lower spatial resolution but a greater temporal scale the European Space Agency (ESA) Climate Change Initiative (CCI) land cover classification (Copernicus Climate Change Service, Climate data store) 300m resolution maps could be used to estimate of the land cover change from 1992 to 2020.

Recommendation

o Our recommendation is that this study be split into three or four publications:

- i. Mapping of wetlands in Africa (and possibly relate these to topography or position within the basin);
- ii. Techniques used to differentiate wetlands from drylands and to distinguish between different wetland types (the classes used need to be defined);
- iii. Human settlement patterns relative to wetlands – and possibly some subsets to show trends as human populations increase;
- iv. Carbon storage and balance.

Reviewer #3

(Remarks to the Author)

Version 1:

Reviewer comments:

Reviewer #1

(Remarks to the Author)

Thank you once again for considering my remarks to enhance this study.

While the authors have adequately addressed the proposed comments, I believe there is still room for improvement in terms of results and identifying a clear research question. The issue of coupling fragmentation and wetland typology persists. There is a diversity within the fragmentation of classes in the observed changes. How did you isolate natural changes from those induced by the population?

Another question is whether population density alone can explain these changes. Are there not areas with low density where significant effects are observed, independent of population?

It would be beneficial to include in the discussion or conclusion the limitations of the approach used and potential avenues for future research. I maintain that the study could be divided into two themes, as the two parts investigated are not fully interconnected.

Integrating images at 10m resolution does not inherently make the study more important or useful than others using lower resolutions. The value lies in the accessibility of these images and the potential for similar approaches to be tested with 30m and 100m resolution data, likely yielding comparable results.

Regarding the separability of wetland classes or types, which are more susceptible to anthropogenic pressure?

Given my familiarity with the densely populated Great African Lakes region, what explains the low fragmentation observed in these areas based on your results? The same can be asked for the Nile delta and the Ngiri Ntuma peatland region (between DRC and R Congo), where a very low population density exists. Would the results differ if a polygon-based approach were used instead of a pixel-based one? Please confirm whether African mangroves are typically organic, organo-mineral, or more mineral-dominant.

Why was elevation not considered, as high peatlands and low peatlands exhibit distinct behaviors? Similar considerations apply to other wetland types like Indonesian valleys and plains. Justify the inclusion of the "deep water" class and why others were categorized as marshes.

Topographic indices could provide additional insights. Would you consider creating a new index based on the difference between the NDVI of wet and dry periods?

Ultimately, I believe the article is not suitable for publication in its current form but could be divided into two separate articles with distinct themes.

Other comments are in the pdf file.

Kind regards

Reviewer #4

(Remarks to the Author)

The manuscript deals with important topics around extent, fragmentation and degradation, and carbon storage in African wetlands and includes an impressive amount of work. The study is remote sensing based and aims at a continental scale, while considering different climatic zones in the continent. The study is without doubt useful and necessary due to the importance of wetlands for ecosystem functions, including the global carbon cycle. The newly applied/modified methods aim at providing more accurate estimates of wetland extent than previous works, due to a finer scale that includes smaller wetlands. Also, the relations between wetland fragmentation and population density are a major aspect of the work, by suggesting a Wetland Fragmentation Population Index, which may help to assess, understand, monitor, and predict wetland fragmentation / degradation on the African continent, as well as resulting greenhouse gas emissions.

However, in the current version, I see both needs and potentials for improving the manuscript before publication. Some aspects remain unclear, and while I am sure the authors thought about them, they need to be better clarified and discussed in the presented work.

Main comments / suggestions:

The Objectives: While the main objectives are mentioned in the introduction, I would recommend the authors to clearly state all the objectives of the work at the end of the introduction in a more concise way for an easier understanding, e.g. as listing them as opposed to running text. The authors might as well consider reducing the number of objectives and focusing on key objectives / or potentially make two different publications.

The introduction currently has some details on the methods which might not be necessary in the introduction (rather in the method section) (66 ff and 74 ff).

On the other hand, the text should be clearer on the definitions of wetland fragmentation and wetland degradation concerning the differences and overlaps between these terms / concepts.

Wetland fragmentation, as the authors state themselves with reference to the Congo Basin, is not necessarily caused by human activity, but can be simply attributed to topography, even in densely populated areas. Wetland degradation on the other hand, does not necessarily mean wetland fragmentation. Wetlands can be intensively used for (dry season) cropping, which can strongly reduce their ecosystem service provision and functions, but still be a wetland. Wetland fragmentation as a process would imply to me a loss of the wetland status, by altering the hydrology (drainage, dams, etc.).

A better definition / explanation of the wetland types would also be helpful. While it is included in the supplementary material, it would also be helpful to get more information in the text.

The aspect of land use does not play a big role in the manuscript, even though many wetlands are agriculturally used, and agriculture is described as a main driver for wetland degradation. The method section on Carbon loss estimations speaks of land use categories (line 447) that are not mentioned elsewhere in the document. If these categories have been assessed, it would be valuable to indicate their extent / proportions in the result section, as well as to explain which land use classes were used at all in the methods. If land use was not assessed in the work, I recommend this to be stated as a limitation in the discussion.

The language and formal aspects need a thorough check and revision. E.g., some parts of the methods are written in present tense and others in past tense, without an obvious reason for the use of different tenses. Sometimes a space between number and unit is missing, and between numbers and hyphens, eg line 19, 24,70,85 ...

Specific comments / suggestions:

Line 32: "some of" not necessary

Line 34: "endemic" not necessary here, many wetland species are cosmopolitan or pantropic species

Line 43: what kind of development activities? Aren't they included in the previously mentioned pressures?

Line 54: significant in which sense?

Line 63: this sentence should be rephrased.

Line 95 ff: The numbers and percentages do not seem to match, or it is unclear to me, what the reference is. 436.000 km²

are more than 33% of 947.000 km².

Line 116ff: The classification is based on Köppen-Geiger, according to the reference. This should be mentioned in the text.

Line 127: it should be TWD instead of TW. Again, what does significant mean here? Higher? Or is it a result of a statistical test?

Figure 2 b): in greyscale printing, only 4 classes are visible. The colors / shadings should be adapted.

Line 170 ff: this is an important aspect. Please also mention it in the discussion

Line 178: How specific are these values? Are they specific for climatic zones in Africa? Please be clearer about this.

Table 1: Please add a reference year for the population.

Line 209: it should be WFPI

Line 232ff. This should be part of the results as well.

Line 306: What is the purpose to get the population data for all these years?

Line 311ff: according to line 116, the classification is only climate based, here it includes ecological aspects. Please clarify.

Line 326: Introduce RF as an abbreviation here.

Line 328: I wonder if these details are relevant for this publication, a reference for the method might justify shortening this paragraph.

Line 353: Abbreviation TWD missing.

Table 2: Why was a mean value used in TWD? Shouldn't there be season specific values?

Version 2:

Reviewer comments:

Reviewer #1

(Remarks to the Author)

Thank you for your feedback, comments, and responses to the questions aimed at clarifying and improving the quality of your article. I have carefully reviewed the work, which is well-written, but I still find it somewhat limited and less impactful for publication in the journal. I therefore suggest that the authors submit the article to a journal that focuses on GIS-related topics instead. For a potential publication in Nature Communications, in my humble opinion, the study should present a more compelling argument on the fragmentation of wetlands associated with high population densities across Africa. The justification for the choice of image type to enhance the understanding of small wetlands is discussed, but does it also allow for the same level of understanding regarding fragmentation? The resolution of population-related information also raises concerns. The justification for fragmentation in previously mentioned regions (such as the Great Lakes region, the Nile Delta, etc.) is also questionable. I am also not sure about the finding that marshes are superior to swamps. Many studies show the opposite result. Regarding the results, a very high level of fragmentation in the Congo Basin is reported, but I remain doubtful about these findings, especially in a study linking this to population density. The same concern applies to the Sudd. The main message of the article is the fragmentation of wetlands due to population density, but this cannot be generalized across the entire continent, as there are confusions, overestimations, or underestimations in a majority of regions.

Reviewer #4

(Remarks to the Author)

The authors addressed all my comments from the previous review, and I am generally satisfied with their responses and modifications.

However, I would still suggest a few minor revisions in the introduction section before publication:

1. Lines 97: While the authors have addressed my comment on natural wetland fragmentation, they clarify that this may be due to natural processes.

However, I still think that the fact that wetlands can naturally occur in fragments due to topography has to be made more clear. It has been done e.g. in the discussion (e.g. lines 225) but is lacking in the introduction. It can be done easily, e.g., by adding in line 97: "Besides topographic reasons for the occurrence small wetland patches in a landscape, formerly larger wetlands can become divided or separated into smaller patches.."

2. The reasons for fragmentation mentioned in lines 98 ff, and 109 ff are almost the same words. Here the text could be simplified and repetitions could be avoided.

3. The objectives are better stated now. As a further suggestion, an introductory sentence before the objectives would improve the smooth flow of the text.

Response to review comments

Reviewer #1 (Remarks to the Author):

Thank you for considering me as an reviewer for this study on the large-scale fragmentation of wetlands across the African continent. The work presents significant content on the theme of wetland protection. Overall, this study provides important results, with well-written paragraphs and a logical flow. However, in my humble opinion, the new information provided still requires significant changes and additions. This raises questions about its suitability for publication in a journal like Nature Communications. Therefore, I propose major revisions to the article, with the possibility of submission to another Nature journal, such as Scientific Reports, for example. You will find my comment in the attached file.

Response: Thank you for your positive comments and useful feedback

1) Line 11-12: I am not sure that high-resolution map of Africa is lacking (Zhang et al., 2024, Yan and Niu, 2009; Anzhen Li et al., 2022, etc. have studied that question).

Response: Our study uses Sentinel-2 imagery as our primary data source, which has major advantages over Landsat/Modis used in the studies mentioned by the reviewer. The spatial resolution of Sentinel-2 imagery is 10m, relative to 30m for LandSAT and 250m for MODIS. Sentinel-2 (2015-2022) imagery also provides the most up to date snapshot of wetland extent. In addition to the generation of this improved high-resolution map, our study also goes on to use it to study wetland fragmentation across the continent and make estimates of carbon stores and fluxes.

2) Line 15: Could you please add one or two sentences on the methodological approach used in the paper?

Response: Sentences about the methodological approach have been added in Line 15 (Page 1), as follows; “We use indices derived from seasonal composite images and a random forest classification algorithm to classify wetlands in Africa. We use the resulting 10m wetland map and datasets of human population to investigate the relationship between wetland fragmentation and population”.

3) Line 20: modify the sentence please

Response: The sentence has been modified in Line 23 page 1 as follows “We estimated the total carbon stock of African wetlands to be 54 ± 11 Gt, larger than that of Europe (12-31 Gt)”.

4) Line 21: how did you find and write the value of C carbon for European wetlands? (12-31 Gt)

Response: The value for European wetland carbon was sourced from reference 34 in line 180-183 (Page 8), as follows:“Our new continental map indicates that African wetland contains 54 ± 11 Gt of carbon which is around 5% to 9% of wetland soil carbon stored globally ($520 - 710$ Gt C)²⁰, and higher than that of European wetlands (12-31 Gt)³⁴”.

5) Line 39: more other activities are mentioned In African wetlands as highlighted in some papers, could you please revise this please

Response: More human activities that impact wetlands in Africa have been added (line 40 - 44) page 2 as follows; “Wetlands in Africa are experiencing immense pressure from human activities, the most

important being direct drainage and conversion to farmland, diversion of water away from wetlands for agricultural irrigation, population growth and urban expansion into wetland areas, pollution, overgrazing, and hydropower development”.

6) Line 57: missing information on why studying *wetland fragmentation*? Wetland fragmentation is a significant threat to the health and functioning of wetland ecosystems and highlights the importance of conservation efforts aimed at preserving and restoring these valuable habitats. Wetlands become divided or separated into smaller, isolated patches or fragments. This fragmentation can occur due to various human activities and natural processes, such as urbanization, agriculture, infrastructure development, and changes in hydrology. Did you consider that? OR highlight that in the introduction please

Response: Information on wetland fragmentation has been added in Line 60-65 (Page 2) as follows: “Wetlands can become divided or separated into smaller, isolated patches or fragments due to both human activities and natural processes, including urbanization, agriculture, infrastructure development, and changes in hydrology. Wetland fragmentation is a significant threat to the health and functioning of wetland ecosystems and highlights the importance of conservation efforts aimed at preserving and restoring these valuable habitats”.

7) Line 60-63: To be revised

Response: We have revised the sentence in line 81-84 (Page 3) as follows; “We analyse the relationship between wetland patchiness derived from our map and population data from Gridded Population of the World database (GPW V4) and test the hypothesis that highly fragmented wetlands are associated with large populations”.

8) Line 82: Nothing in the south? South Africa, Zambia, Zimbabwe, Botswana, Tanzania, Tchad, Morocco, etc. What are you calling wetland complexes?

Response: There are wetlands in these areas and the high-resolution map shows them. To highlight this further in our article additional plates have been added to Figure 1 to show zoomed in locations of

wetlands in southern Africa, Chad and Morocco. We refer to a wetland complex as an area consisting of two or more wetland types. We have now added this explanation to the paper at line 100 (Page 4), as follows “...wetland complexes (regions where there are two or more wetland types clustered together) ...”.

9) Line 58: please specify the number of ground control for each wetland type and distribution. Specify, the source of the data map of climatic zones, the process used to estimate the carbon stock,

Response: We added the number of ground control points for each wetland (Line 75-76, page 3), “Control points were grouped into five wetland types including marsh (2202), mangrove (1477), swamps (1891), peatland (1580) and seasonal wetland (1054)”. The data source for climatic zone classification is reference 29 (Line 116, page 5), “We present the spatial distribution of different wetland types based on our high resolution mapping, according to the five main climatic regions in Africa ²⁹”, and the process of carbon stock estimation has been explained (Line 87-89, page 3), “We calculate total carbon stocks for each wetland type by multiplying the total area of the wetland with typical values of carbon stock per hectare estimated by previous studies²²⁻²⁸”.

10) Line 139-140: please specify and be clearer please. All the values from 156 to 177 have to be justified and sourced.

Response: The revised statement is in current line 167-173(Page 8), “However, large wetland areas with a high concentration of fragments (for example the Congo basin wetlands) that are far away from settlements or sparsely populated show little or no relation between fragmentation and human populations (Figure 3). The high concentration of fragments in the Congo basin are thought to be geomorphologically and climatologically controlled rather than driven by human activities²⁵, though these peatlands could be highly sensitive to human-induced fragmentation in the future”. Here, we point out that our model shows a strong relationship between high population density and wetland fragmentation, though some highly fragmented wetland areas (Figure S2) do not coincide with high population. All values are calculated using the emission factor and carbon stock value for each wetland

type obtained from the reference cited in line 87 page 3.

11) There still more things to add in this discussion please.

Response

We have added more details to the discussion section at

- Line 229 (page 10), “Our study shows close similarities with smaller geographical scale studies, such as the peatland map of Angolan highlands constructed by Lourenco et al. 2022 (see supplementary figure S6)”.

-Line 250 (page 11) “Our future work will concentrate on gathering more and better-quality ground control data to support future timeseries analyses”.

Line 252 (page 11) “We explored the possible impact of African wetlands on global climate through net carbon uptake/loss under natural and a range of drained conditions. We found that the three selected wetland types (peatland, mangrove and marsh) under drained conditions could contribute up to 3% of global net annual carbon loss, a value that might be much higher if data for emissions from other wetlands become available and included in the estimation. Human activities have been widely reported to be a key driver of wetland degradation”

12) Still confusion on how you certified the value used. You may have chosen a date (2015), for example, because before this date, some older databases may include wetland areas that are no longer considered as such (many marshes, for example, have disappeared). Other wetland areas are so small and insignificant that they are not represented. Additionally, in some conflict-affected regions where it is difficult to obtain field data, there are limitations in these areas.

Response: We have added sentences to make the text clearer at Line 68-74 (page 2) “We verify the classification of each wetland control point by visual interpretation of Digital Globe very high spatial resolution images (< 1 m pixels) through Google Earth and spanning 2018 -2021. Our independent sources of data for control points for wetland types include papers, reports, and academic theses from

various dates spanning 2015 - 2019, so we consider verification with the more recent Digital Globe imagery (2018 - 2021) a necessary step for classification of the 2020-2021 Sentinel 1 and 2 imageries”.

13) Line 246: Please specify how you find 16 spectral bands of 10m resolution for Sentinel 2. Specify also which type of Sentinel you used: 2A or 2B, etc.

Response: Sentinel 2 has 13 spectral bands - we have rectified the typo from 16 to 13 spectral bands in Line 294 (page 12). We use sentinel 2A and 2B data, as stated in the article at line 294.

14) Line 265: Here, I process.....

Response: We have rephrased the sentence in line 313 (page 13) as follows; “Here we process the images collected from Sentinel-1 and -2 images for the period of 2020-2021 to develop optical and radar indices for each climate zone”.

15) Globally the method is not clear, how you integrated and maintained the 10m resolution, how you combined the population and sentinel images with different resolutions.

Response: Sentinel 1 GRD (Interferometric Wide swath mode (IW) has a pixel spacing of 10m and Sentinel 2 imagery has a 10 m resolution (Line 290-294). However, for our analysis we use only the count of wetland fragments estimated at a resolution of 10 km to allow comparison to the gridded population data at 10 km resolution. This resolution was selected because it was found to be the mean dimension of wetland fragments from our earlier study (Garba et al., 2023) (Line 384-388).

16) Overall, this approach could not identify the inland valleys, floodplains, ponds, etc. wetlands. The seasonal wetlands also cannot be identified.

Response: This is not true. We categorize these features based on the type of wetland cover as identified from optical and radar imagery. For example, the floodplains in the Niger delta of Nigeria are covered by different wetland types such as swamps and marshes. The zoomed areas in Figure 1 display some areas covered by seasonal wetlands, for example seasonal wetlands in Morocco (Figure 1f) identified by using composites showing maximum wetness.

17) Since wetlands are defined based on topography, hydrology, vegetation cover, etc. Why did you select only vegetation cover? Using Optical data from sentinel? Why did not use the DTM, DEM images with topographic indices such as slope, curvature, TWI, TPI, etc.

Response: We use 8204 ground control points with detailed information about wetland type and location, giving us confidence that our classification of optical and radar imagery is acceptably accurate even in the absence of topographic data. We also limit our classification to wetland classes that we are confident can be distinguished from optical and radar imagery alone. However, we intend to further refine the models in a future paper by including topographic variables. This will require choices about data resolution, since there is currently no freely available, uniform, global DEM at as high a resolution as Sentinels1 and 2.

18) Please justify also why choosing the RF model that others. Why did not you use other accuracy coefficients such as AUROC, TSS, DeLong Test, etc.

Response: This has been added to line 329 – 333 (Page 13) “RF is more robust compared to other classification algorithms, solving the problems of over-fitting with other decision trees. The RF is particularly suitable for handling variation within land cover classes and reducing noise in the data. It does not require prior knowledge of the data distribution compared to other classifiers”.

19) Other factors such as distance to villages, city, road, etc. should be used to better highlight the population activities that are drivers of change in wetlands.

Response: We choose to use gridded population data to ease comparison with wetland fragmentation. This approach has the advantage of using a relatively uniform dataset instead of relying on population indicators (like distances to villages, city, road) that vary from country to country. Deriving a novel set of meaningful population indicators for comparison would be a substantial additional piece of work, and will be a focus of future efforts.

20) Line 365 to 400: Not clear, need to be as clear as possible.

Response: We have added sentences to make the text clearer in line 438-442 (page 18) “An emission/removal factor is a coefficient that quantifies the emissions or removals of a gas per unit area.

It is calculated based on measurement of samples, averaged to develop a representative rate of emission for a given activity level under a given set of operating conditions”.

21) Some sentence still need grammar check.

Response: We have reviewed all grammar and spelling mistakes

Reviewer #2 (Remarks to the Author):

This paper addresses the lack of high-resolution data on Africa’s wetlands - the extent, the intactness of the wetlands and carbon storage in the wetlands. There is an abundance of information in this paper. Using this wetland map, and the GPW V4 data used to map of human densities, the authors have derived a measure of wetland intactness/fragmentation and the volumes of carbon stored in the Wetlands of Africa.

In our opinion, the highlight of this paper is the production of a 10m resolution map of Africa’s wetlands. This is a valuable output which is likely to improve on previous maps with lower resolutions that cannot pick up all the smaller wetlands.

Response: Thank you for your positive comments and useful feedback.

General comments/questions:

1) The paper does not have a specific question to be answered or hypothesis to be tested.

Response: We state a hypothesis that highly fragmented wetlands are associated with large populations in Line 82 – 83 (Page 3).

2) It is a multi-disciplinary study – involving computer techniques, cartography, quantification of carbon etc. However, the disciplines do not link seamlessly. It seems as if there are different degrees of rigor in assessing accuracy and precision at different levels in this study. For instance, the wetlands are mapped at a 10 m resolution, but the human distribution patterns at a 10 km resolution. The issues in bringing together these resolutions are not adequately dealt with. There needs to be some understanding of where humans settle. Empirically their distribution is a narrow linear pattern along rivers and in floodplains

which is difficult to evaluate when using a 10 km grid for human distribution. It would be good to give some analysis of this.

Response: We used a 10 km grid for fragmentation analysis based on our previous studies that suggested that average continuous wetland patches cover an area of 10 -11km² (Garba et al., 2023) (Line 83-85, page 3). The wetland fragmentation uses the total count of wetland fragments, using the 10 m data, within a 10 km grid used as input in the model. The gridded population data is also down sampled from 1km to 10km cells to be able to be accurately overlaid with the wetland patches. We have clarified these points in the text and revised the caption for Figure 3 to help avoid confusion in Line 151-153(page 7) “Data for wetland cover uses our 10 m gridded dataset to determine fragmentation within each 10 km cell. Data for population density uses 1 km data reprofiled to 10 km”. A common issue with the gridded population data is misallocation of population to areas outside the urban areas. However, the gridded data does allow a fair, first order comparison to datasets derived from EO data.

3) We also have a concern that there may be a compounding of errors – from population censuses, to allocation of densities in 10 km grids, and errors relating to overlapping these on wetlands mapped at a small scale. This is something that does need to be addressed.

Response: We have clarified in the manuscript that we use the full resolution EO data to make estimations of fragmentation at a lower resolution and compared that to population data downsampled to the same cell size at Line 151-153 “Data for wetland cover uses our 10 m gridded dataset to determine fragmentation within each 10 km cell. Data for population density uses 1 km data reprofiled to 10 km”

Line 384 “We compared the distribution of wetland fragments and population at the same cell size across 10 km grid areas”. This approach minimises errors associated with comparing at differing resolutions, although of course it limits the spatial scale over which we are able to make conclusions. We down sampled the 1 km grid population taking the average population within 10 km grid minimizing errors from small scale grid comparison (Line 379-381, page 16). We have larger uncertainties in our fragmentation (see line 376-378, page 16) compared to the population data, although we acknowledge that the uncertainty in populations datasets will vary geographically.

4) Overall, the study would benefit by having a discussion on the processes involved – in wetland fragmentation and human settlement patterns. And then, what are the global impacts of such fragmentation.

Response: A discussion on wetland fragmentation has been added in line 60-65 (Page 2) “Wetlands can become divided or separated into smaller, isolated patches or fragments due to both human activities and natural processes, including urbanization, agriculture, infrastructure development, and changes in hydrology. Wetland fragmentation is a significant threat to the health and functioning of wetland ecosystems and highlights the importance of conservation efforts aimed at preserving and restoring these valuable habitats”.

Mapping of wetlands

5) With the mapping of wetlands, we question the classification of the wetlands into the categories the authors have used. There needs to be a clear definition of each wetland type. What is the difference between swamp and marsh? Swamp is usually used to describe a wetland with trees while marsh is shorter vegetation. Papyrus wetlands are a feature of large areas in Africa – are they swamp or marsh? And what distinguishes wooded wetland from seasonal wetland? And peatlands from marsh? How is a ‘seasonal wetland’ determined? The term ‘wetland complex’ is used – what is meant by this?

Response: We have added a clear description of each wetland type that we have used in supplementary Table S2. These classes capture critical differences in wetland vegetation, soils and water levels (see Supplementary Table S3-S7) and are separable using optical and radar-derived indices from freely available satellite datasets. Seasonal wetlands were identified using the maximum value from our seasonal composites (line 351-352, page 14). We refer to a wetland complex as an area consisting of two or more wetland types in Line 100 (page 4) “regions where there are two or more wetland types clustered together”.

Table S2 Description of each wetland type used in this study.

Wetland class	Soil systems	Water sources	Typical settings and features	Plant species
Marsh	Mineral	Direct flow from lakes, streams, precipitation	Edges of lakes and streams, Coastal zone (salt/tidal marshes)	Herbaceous
Swamp	Mineral or Organic	Precipitation, groundwater, freshwater flooding from rivers or lakes	Along large rivers or on the shores of large lakes	Woody, forested
Mangrove	Organic	Precipitation, groundwater and tidal flow.	Coastal zone mostly grows in sheltered low lying coasts estuaries, and lagoons	Trees and shrubs
Peatland	Organic	Groundwater inflow or precipitation	Standing water of lakes or margins of slow flowing rivers,	Herbaceous plants, Shrubs, small trees.
Seasonal wetland	Organic or mineral	Precipitation	Low lying areas and open fields.	Herbaceous

6) Verification: How does the wetland extent compare to other wetland maps generated for specific regions in Africa with smaller geographical extents e.g. Lourenco et al. 2022 mapped the Angolan highlands peatlands. How does this (or other similar maps) compare with the mapping produced for this study? For this study the verification is at a density of one point per 200 km². Is this adequate for such a fine-scale resolution map covering such variable conditions as the whole of Africa?

Response: We have added a figure comparing our study with Lourenco et al. 2022 in the supplementary document (Figure S6). Line 2290 (page 10), “Our study shows close similarities with smaller geographical scale studies, such as the peatland map of Angolan highlands constructed by Lourenco et al. 2022 (see supplementary figure S6). We mention the issues about future research benefitting from more ground control data and better carbon characterization for African wetlands in line 251, “Our

future work will be concentrated on gathering more and better-quality ground control data to support some future timeseries analyses”.

Mapping of people

7) People do live where there is water and wetlands. This is especially the case in regions where water is not abundant. So, one would expect a correlation between human numbers and wetlands. Has there been any change in this over time? Or is it just that, as human numbers increase, so this effect is more pronounced. This can only be shown by having a time series – to detect if this human distribution pattern stays the same or changes over time, and is there a corresponding change in wetland fragmentation?

Response: Our comparison shows that there is high correlation between human numbers and wetlands. We will explore time series changes of wetland fragmentation/ population in further studies. However, a high spatial resolution time series analysis of wetlands and wetland change is out of scope for the current paper and would be a significant new body of work.

Fragmentation

8) There are causes of wetland fragmentation that do not relate to human population density that need to be discussed? These include the downstream impacts of dams, the impacts of lowering of groundwater table, the impacts of mining and large-scale agriculture.

Response: We have added these points in Line 60-65 “Wetlands can become divided or separated into smaller, isolated patches or fragments due to both human activities and natural processes, including urbanization, agriculture, infrastructure development, and changes in hydrology”.

We also highlight an example from the results of our study in Line 167-173 “However, large wetland areas with a high concentration of fragments (for example the Congo basin wetlands) that are far away from settlements or sparsely populated show little or no relation between fragmentation and human populations (Figure 3). The high concentration of fragments in the Congo basin are thought to be

geomorphologically and climatologically controlled rather than driven by human activities²⁵, though these peatlands could be highly sensitive to human-induced fragmentation in the future”.

9) To separate natural fragmentation patterns (e.g. due to the nature of topography) from human-caused changes it is necessary to show the relationship over time between numbers of people and a change in fragmentation/transformation. Sentinel 2 was fully commissioned by 2017 – is it within the scope of this study to compare wetland extent between 2017 and 2020? Or at a lower spatial resolution but a greater temporal scale the European Space Agency (ESA) Climate Change Initiative (CCI) land cover classification (Copernicus Climate Change Service, Climate data store) 300m resolution maps could be used to estimate of the land cover change from 1992 to 2020.

Response: It would be a significant new undertaking, and essentially an entirely new paper, to explore a 10 m resolution timeseries analysis (e.g. requiring timeseries of ground truth data). We aim to explore temporal changes of wetland fragmentation/ population in our future papers.

Recommendation

10) Our recommendation is that this study be split into three or four publications:

- i. Mapping of wetlands in Africa (and possibly relate these to topography or position within the basin);
- ii. Techniques used to differentiate wetlands from drylands and to distinguish between different wetland types (the classes used need to be defined);

- iii. Human settlement patterns relative to wetlands – and possibly some subsets to show trends as human populations increase;

- iv. Carbon storage and balance.

Response: Thank you for the recommendations. We think that our current paper is a significant new body of work which incorporates most of these elements already as one powerful and impactful study. We therefore strongly prefer to keep the paper as one body of work rather than split it into smaller papers. Future papers will aim to incorporate time series analysis and topographic variables.

Reviewer #3 (Remarks to the Author):

Response: That's great – we very much appreciate your feedback.

Response to review comments

Reviewer #1:

We thank the reviewer for their helpful guidance and feedback

1) There is a diversity within the fragmentation of classes in the observed changes. How did you isolate natural changes from those induced by the population?

Response: We assess the relationship between wetland fragmentation and population density using a fuzzy membership approach. While wetlands exhibiting high fragmentation and located near densely populated areas may indicate pressure from human activities, this correlation does not prove causality. Processes that can cause fragmentation of wetlands include urbanization, agricultural expansion, water management activities such as flow diversions and channelization, deforestation, and pollution. We discussed these at line 97. The intersection of fragmentation and high population density is also indicative of increased future vulnerability to human activities, as discussed at line 293 “The WFPI analysis generally identifies sites where the impact of human activities results in patchiness of the surrounding wetlands. The overlap between fragmentation and high population density indicates an increased susceptibility to future human impacts”.

2) Another question is whether population density alone can explain these changes. Are there not areas with low density where significant effects are observed, independent of population

Response: We prioritized the assessment of human-induced wetland fragmentation because human activities are the leading cause of rapid and extensive ecosystem disruption, particularly in densely populated or rapidly developing areas (Vitousek et al., 1997; Foley et al., 2005; Pörtner et al., 2022). In contrast, natural fragmentation processes, such as geomorphological patterns or hydrological dynamics (e.g., those observed in the Congo Basin wetlands as already discussed in the manuscript at Line 210), typically occur over extended timescales and are less directly impacted by short-term policy or management measures.

3) Integrating images at 10m resolution does not inherently make the study more important or useful than others using lower resolutions. The value lies in the accessibility of these images and the potential for similar approaches to be tested with 30m and 100m resolution data, likely yielding comparable results

Response: Lower-resolution data (e.g., 30m Landsat or 100m MODIS) may yield similar results for larger wetlands or general trends but may fail to capture fine-scale variability and smaller fragments. The lower the resolution, the greater the impact of 'mixed pixels' where wetland features are averaged with non-wetland features. This can obscure the presence of small wetlands, leading to underrepresentation or misclassification. Integrating 10m resolution data inherently adds value for studies like ours, focused on identifying small, fragmented wetlands and updating estimates of the significance of African wetlands for global climate. We discuss the specific value of the higher resolution wetland maps to our study at lines 273.

4) Regarding the separability of wetland classes or types, which are more susceptible to anthropogenic pressure?

Response: We have discussed this in line 61 “Amongst these wetland types, marshes are especially susceptible to anthropogenic pressures because of their accessibility, fertile soils, and proximity to densely populated regions. High demand for agricultural and urban development, combined with inadequate protection measures, makes these wetlands highly vulnerable and often heavily exploited.”

5) Given my familiarity with the densely populated Great African Lakes region, what explains the low fragmentation observed in these areas based on your results? The same can be asked for the Nile delta and the Ngiri Ntuma peatland region (between DRC and R Congo), where a very low population density exists. Would the results differ if a polygon-based approach were used instead of a pixel-based one? Please confirm whether African mangroves are typically organic, organo-mineral, or more mineral-dominant.

Response: We have discussed this in line 228 “The low fragmentation in the Great African Lakes region is likely due to the hydrological stability and the presence of large, continuous wetland

systems. These wetlands are naturally resistant to fragmentation, while human activities like agriculture and urbanization are primarily confined to upland areas”.

For areas like the Nile delta and Ngiri Ntuma peatlands (between DRC and R Congo), the high fragmentation despite low population is linked to natural factors such as hydrology and geomorphology as we highlighted in line 295 “It should be noted that extensive wetland regions with a high density of fragments, such as those in the Congo Basin, generally show minimal correlation between fragmentation and human populations, especially in areas distant from settlements or with sparse human activity. The fragmentation in these wetlands is largely attributed to geomorphological and climatological processes rather than anthropogenic influences. However, these peatlands remain susceptible to potential future fragmentation driven by human activities, emphasizing the need for future studies that examine wetland change and detect change in fragmentation to support vigilant conservation and management strategies.”.

African mangroves are generally classified as organo-mineral systems (Spalding et al., 2010; Bunting et al., 2018).

6) Why was elevation not considered, as high peatlands and low peatlands exhibit distinct behaviors? Similar considerations apply to other wetland types like Indonesian valleys and plains. Justify the inclusion of the "deep water" class and why others were categorized as marshes.

Response: We have discussed this in line 308 “We have used detailed ground control points specifying wetland types and locations, ensuring that incorporating topographic indices would not substantially affect classification accuracy. However, we aim to refine the models further in future research by integrating topographic variables to enhance their precision and applicability”.

7) Topographic indices could provide additional insights. Would you consider creating a new index based on the difference between the NDVI of wet and dry periods?

Response: We have used both wet and dry periods for classification of some the climate zones as discussed in line 425 “Seasonal wetlands were also identified using the maximum value from our seasonal composites of wet and dry season in each climate zone. For the Tropical Wet and Dry

and Mediterranean subtropical climate (MED) zones, the variables constructed for the wet and dry season from the mean pixel value composites were used to train the RF classifier”

8) Ultimately, I believe the article is not suitable for publication in its current form but could be divided into two separate articles with distinct themes.

Response: Thank you for the suggestions. We believe our current paper represents a substantial and cohesive body of work, integrating many of these elements into a single, impactful study. As such, we prefer to maintain it as one comprehensive piece rather than divide it into smaller publications.

9) Please confirm that area occupied by marshes is larger than swamps

Response: Yes, the area of marshes is greater than swamps as discussed in line 145 “Marshes, covering 436,743 km² (46% of total wetlands), are more extensive than swamps, which account for 231,776 km² (24%).”

10) Some marshes in South and central Africa are seasonal

Response: We classify seasonal marshes in this region under the broader category of marshes, emphasizing vegetation type as the primary distinguishing factor, as stated in line 67.

11) Is the difference based on the fragmentation or covered areas? I think it's normal to have high total carbon stock in Africa than Europe (surface, types, etc)

Response: The higher total carbon stock in Africa compared to Europe is primarily based on differences in wetland area, as stated at line 238, “ Our new continental map indicates that African wetland contains 54 ± 11 Gt of carbon which is around 5% to 9% of wetland soil carbon stored globally (520 - 710 Gt C)²⁰, and higher than that of European wetlands (12-31 Gt)³⁴ based on differences in wetland area”.

12) Please add a source here. As globally 70*30 is mostly used.

Response: We have added the source in line 348 “Thus, the control points were grouped into an equal number of training and testing points to ensure robust accuracy assessment (Mahdianpari et al., 2018; Zhang et al., 2023)”.

13) I think you did not use all the bands; please specify the one you used.

Response: We have specified the bands used in lines 361 “We use blue (0.496 μm , band 2), green (0.560 μm band 3), red (0.665 μm , band 4), and near infrared (NIR, 0.835 μm , band 8), and short-wave infrared 2 (SWIR2 2.202 μm , band 12) bands”.

14) Why did you therefore select the images of the same year to reduce error

Response: We discuss our selection of imagery for training and classification in line 343. Given that the Sentinels were only launched in 2015/16, using the most recent imagery was prioritized to ensure that the analysis reflects current wetland dynamics and conditions. Wetland ecosystems are highly dynamic, and recent changes driven by human activities or climate impacts are better captured with up-to-date satellite imagery.

15) Please give more details how this was made

Response: The details of how the population was estimated for the target years is available documentation for the Gridded Population of the World, Version 4 (GPWv4) accessed through: <http://sedac.ciesin.columbia.edu/data/collection/gpw-v4/methods/method1>. This is referenced in our article at line 363.

16) I don't remember seeing those numbers for each climate zone

Response: The number for each climate zone has been given in supplementary material Table S3 – S7. We refer to this in the main text at line 385 “TWD consists of 3,120 control points, TW includes 2,550, SARD has 1,144, ARD contains 848, and MED comprises 536”.

17) Results from this can be added as Supplementary materials

Response: We have added the result of variable importance to supplementary materials in figure S7.

18) Results will depend on the tableau of number of controled points for each wetland type.

How did you manage training points with confusion of wetland types

Response: To manage confusion in training points across wetland types, a systematic approach was implemented. High-resolution imagery verification ensured accuracy in control point labeling. Class-specific spectral and radar features were used to distinguish overlapping wetland types, and performance was evaluated by calculating overall accuracy, user’s and producer’s accuracy, and the kappa coefficient based on a confusion matrix derived from validation data as discussed at line 440.

19) I suppose that the variables varied from on wetland type to another. So how did you manage that?

Response: This has been clarified in line 401. We compute the variable importance five times before selecting the variables that display highest performance for all of the wetland classes.

20) I do not understand how seasonality was analyzed; As it varied from one climate region to another, from one agroecological zone to another

PLEASE CLARIFY THIS

I think this can be done if images where selected again based on seasonality in each region

Response: We selected seasonal images to construct our seasonal composite as we stated in line 419, “To accurately classify the wetland types based on their distinctive features in a particular region, the input variables were extracted from composite images constructed from different pixel values over a particular period of the year. In the arid and semi-arid region, seasonality is often a key property of wetlands. We therefore use the variables extracted from seasonal composites of maximum pixel value to train the RF classifier (Table 2). Seasonal wetlands were also identified using the maximum value from our seasonal composites of wet and dry season in each climate zone.”.

Images were selected depending on the distinct seasonal patterns for each climate region. There is considerable variation in the timing and intensity of wet and dry seasons between climate zones, and so the selection of images was tailored to reflect local conditions. For instance, in the TWD region, where there is distinct rainfall and a dry period within the year, wet season images correspond to peak rainfall months, while dry season images reflect periods of minimal precipitation.

21) What value for you be identified as HIGHEST IMPORTANCE?????

Response: We have discussed this in line 397 “The highest importance is typically placed on the variables that contribute the most to reducing the model's impurity or error. We identify these variables by computing variable importance in an RF algorithm using the Mean Decrease Impurity (MDI) which measures how much each variable improves the model's predictive performance. We run the variable importance algorithm multiple times before finally selecting the variables with higher reduction in impurity considered as higher importance”.

22) WHY DID NOT YOU USED OTHER ACCURACY COEFFICIENT?????

Response: We have explained this in line 444 “We selected kappa coefficient in this analysis due to its widespread application and interpretability in evaluating agreement between observed and predicted classes, making it a reliable metric for wetland classification studies”.

23) As most of population are located in or around cities and few in rural areas. This looks like you are assessing the effect on only wetlands located in the vicinity of cities. So why you didn't take access to road, distance from wetland to city as also factors. As mentioned in most articles as fragmentation elements

Response: We considered all wetlands across the study area, encompassing urban, peri-urban, and rural regions, rather than focusing solely on wetlands near cities. While it might appear that the analysis primarily reflects urban impacts due to population concentration near cities, the study captures a broader picture of wetland fragmentation and population distribution.

24) This was made at 10m, so please specify the algorithm used to scale these grid areas please

Response: This is explained at line 487 “We use an algorithm similar to spatial aggregation by overlaying 10 km x 10 km grid on the original 10m resolution fragmentation map. For every 10m cell within a 10 km grid, the number of unique wetland fragments is calculated. We then identify and label distinct fragments within each grid. The fragment count within each 10 km grid cell is computed to derive a metrics of the total number of fragments”.

25) Is it 1km or 10km? in the methodology it is said to be 1km. Please confirm this

Response: We use 10 km grids.

26) Why two??? and not three?? Low, medium and high

Response: The value is a range from 0 to 1, as referred to in supplementary figure S3.

Reviewer #4 (Remarks to the Author):

The manuscript deals with important topics around extent, fragmentation and degradation, and

carbon storage in African wetlands and includes an impressive amount of work. The study is remote sensing based and aims at a continental scale, while considering different climatic zones in the continent. The study is without doubt useful and necessary due to the importance of wetlands for ecosystem functions, including the global carbon cycle. The newly applied/modified methods aim at providing more accurate estimates of wetland extent than previous works, due to a finer scale that includes smaller wetlands. Also, the relations between wetland fragmentation and population density are a major aspect of the work, by suggesting a Wetland Fragmentation Population Index, which may help to assess, understand, monitor, and predict wetland fragmentation / degradation on the African continent, as well as resulting greenhouse gas emissions. However, in the current version, I see both needs and potentials for improving the manuscript before publication. Some aspects remain unclear, and while I am sure the authors thought about them, they need to be better clarified and discussed in the presented work.

Response: Thank you for your positive comments and useful feedback

Main comments / suggestions:

1) The Objectives: While the main objectives are mentioned in the introduction, I would recommend the authors to clearly state all the objectives of the work at the end of the introduction in a more concise way for an easier understanding, e.g. as listing them as opposed to running text. The authors might as well consider reducing the number of objectives and focusing on key objectives / or potentially make two different publications.

Response: We have listed the objectives in lines 114 “The primary objectives of this study are to: 1) systematically map the spatial distribution of wetlands across Africa by categorizing them into five distinct types—marsh, mangrove, swamp, peatland, and seasonal wetland—based on ecological and hydrological characteristics, 2) analyze wetland patchiness and assess its relationship with population density, testing the hypothesis that wetlands in highly populated areas are more fragmented, 3) estimate the total carbon stocks for each wetland type and calculate potential carbon emissions under pristine and drained conditions across various climate zones”.

2) The introduction currently has some details on the methods which might not be necessary in the introduction (rather in the method section) (66 ff and 74 ff). On the other hand, the text should be

clearer on the definitions of wetland fragmentation and wetland degradation concerning the differences and overlaps between these terms / concepts.

Response: We have clarified the definition of wetland degradation and fragmentation in lines 102 “Wetland degradation refers to a decline in the ecological integrity and functionality of wetlands, characterized by reduced biodiversity, compromised water quality, and diminished carbon sequestration potential. It occurs due to drivers such as pollution, excessive water extraction, encroachment by invasive species, and the impacts of climate change. Unlike fragmentation, which involves the physical separation of wetland areas into smaller patches, degradation refers to the deterioration of the ecosystem’s health and its ability to provide essential services, irrespective of spatial continuity”.

3) Wetland fragmentation, as the authors state themselves with reference to the Congo Basin, is not necessarily caused by human activity, but can be simply attributed to topography, even in densely populated areas. Wetland degradation on the other hand, does not necessarily mean wetland fragmentation. Wetlands can be intensively used for (dry season) cropping, which can strongly reduce their ecosystem service provision and functions, but still be a wetland. Wetland fragmentation as a process would imply to me a loss of the wetland status, by altering the hydrology (drainage, dams, etc.).

Response: We have clarified this in lines 97, “Wetlands can become divided or separated into smaller, isolated patches or fragments due to both human activities and natural processes, including urbanization, agriculture, infrastructure development, and changes in hydrology. Wetland fragmentation poses a serious threat to the health and functionality of wetland ecosystems, highlighting the need for conservation efforts focused on preserving and restoring these valuable habitats. Wetland degradation refers to a decline in the ecological integrity and functionality of wetlands, characterized by reduced biodiversity, compromised water quality, and diminished carbon sequestration potential. It occurs due to drivers such as pollution, excessive water extraction, encroachment by invasive species, and the impacts of climate change. Unlike fragmentation, which involves the physical separation of wetland areas into smaller patches, degradation focuses on the deterioration of the ecosystem’s health and its ability to provide essential services, irrespective of spatial continuity”.

4) A better definition / explanation of the wetland types would also be helpful. While it is included in the supplementary material, it would also be helpful to get more information in the text

Response: We have included a better definition / explanation of the wetland types in lines 40, “Wetlands are dynamic ecosystems that can be categorized based on their hydrology, soil composition, and vegetation types, each supporting unique ecological functions and biodiversity. Marshes, for instance, are wetlands dominated by herbaceous (non-woody) plants, characterized by periodic or continuous flooding. They can be found in both freshwater and saline environments, where nutrient-rich soils foster diverse flora and fauna. Swamps are wetlands with mineral soils (although some classifications also distinguish organic soil peat swamps), dominated by woody vegetation such as trees and shrubs. These ecosystems experience seasonal or permanent flooding and include coastal mangrove swamps, which are crucial for coastal protection, carbon sequestration, and biodiversity conservation.

Peatlands represent a distinct wetland category, defined by the accumulation of partially decomposed organic matter (peat) due to water saturation. Peatlands are further classified into bogs and fens. Bogs are rain-fed (ombrotrophic) systems that are typically acidic and nutrient-poor, often supporting mosses, shrubs, and sometimes trees. Fens, by contrast, are groundwater-fed (minerotrophic) and more nutrient-rich, allowing for a mix of grasses, sedges, and woody vegetation. Seasonal wetlands, another important type, experience periodic inundation during specific times of the year, followed by dry conditions. These include natural systems like ephemeral ponds and human-made systems such as rice paddies, which harbour species adapted to fluctuating water levels”.

5) The aspect of land use does not play a big role in the manuscript, even though many wetlands are agriculturally used, and agriculture is described as a main driver for wetland degradation. The method section on Carbon loss estimations speaks of land use categories (line 447) that are not mentioned elsewhere in the document. If these categories have been assessed, it would be valuable to indicate their extent / proportions in the result section, as well as to explain which land use classes were used at all in the methods. If land use was not assessed in the work, i recommend this to be stated as a limitation in the discussion.

Response: We have added this as limitation in line 328 “Although land use was indirectly considered in the carbon loss estimations, a comprehensive evaluation of its specific impacts was outside the scope of this study. Future investigations should integrate detailed land use data to better understand and quantify its contribution to wetland fragmentation and degradation”.

Specific comments / suggestions:

6) Line 32: “some of” not necessary

Response: We have removed the words “some of”.

7) Line 34: “endemic” not necessary here, many wetland species are cosmopolitan or pantropic species

Response: We have addressed this by removing the word “endemic”.

8) Line 43: what kind of development activities? Aren’t they included in the previously mentioned pressures?

Response: We have removed the “development activities” as the directly or indirectly included in previously mentioned pressures.

9) Line 54: significant in which sense?

Response: We refer to the importance of cumulative coverage of small wetlands to diverse ecological, climatic, and hydrological functions. We have added more text to clarify this in current line 86 “It is therefore not known whether the cumulative coverage of small wetlands is important to diverse ecological, climatic, and hydrological functions and there is a need to ensure appropriate representation of African wetlands for sustainable management and for modelling climate mitigation and biogeochemical cycles”.

10) Line 63: this sentence should be rephrased.

Response: The sentence has been rephrased in lines 96 “Wetland fragmentation poses a serious threat to the health and functionality of wetland ecosystems, highlighting the need for conservation efforts focused on preserving and restoring these valuable habitats”.

11) Line 95 ff: The numbers and percentages do not seem to match, or it is unclear to me, what the reference is. 436.000 km² are more than 33% of 947.000 km².

Response: We have corrected this error in lines 145 “Marshes and swamps are the most dominant wetland covering 436,743 km² (46% of total wetlands) and 231,776 km² (24%) respectively. Peatlands cover 208,842 km² (22%), while seasonal wetlands (5%) and mangroves (3%) have the least coverage”.

12) Line 116ff: The classification is based on Köppen-Geiger, according to the reference. This should be mentioned in the text.

Response: We have added this in the text at lines 168.

13) Line 127: it should be TWD instead of TW. Again, what does significant mean here? Higher? Or is it a result of a statistical test?

Response: We have corrected the typo from “TW” to “TWD” in current line 182. Significant has been replaced with Higher for clarification “Thus, TWD has a higher amount of seasonal wetland cover relative to other climate zones (Figure 2b)”.

14) Figure 2 b): in greyscale printing, only 4 classes are visible. The colors / shadings should be adapted.

Response: The journal publish all materials in full colour so we have not modified the colours.

15) Line 170 ff: this is an important aspect. Please also mention it in the discussion.

Response: We have added this in lines 295 “It should be noted that extensive wetland regions with a high density of fragments, such as those in the Congo Basin, generally show minimal correlation between fragmentation and human populations, especially in areas distant from settlements or with sparse human activity. The fragmentation in these wetlands is largely attributed to geomorphological and climatological processes rather than anthropogenic influences. However, these peatlands remain susceptible to potential future fragmentation driven by human activities, emphasizing the need for vigilant conservation and management strategies”.

16) Line 178: How specific are these values? Are they specific for climatic zones in Africa? Please be clearer about this.

Response: The values are according to climate zones mentioned in current line 237.

17) Table 1: Please add a reference year for the population.

Response: The reference year for the population has been added to Table 1

18) Line 209: it should be WFPI

Response: The typo “WPFI” has been corrected to WFPI in current line 259.

19) Line 232ff. This should be part of the results as well.

Response: The sentence in line 232 has been moved to result section in line 162.

20) Line 306: What is the purpose to get the population data for all these years?

Response: We use the population for 2020. We added text to clarify that in line 378.

21) Line 311ff: according to line 116, the classification is only climate based, here it includes ecological aspects. Please clarify.

Response: Climate zones in Africa differ significantly in both climate and ecological features, resulting in diverse ecosystems across the continent.

22) Line 326: Introduce RF as an abbreviation here.

Response: We have added the abbreviation of RF in current line 399

23) Line 328: I wonder if these details are relevant for this publication, a reference for the method might justify shortening this paragraph.

Response: We have added references to the RF method to shorten this paragraph in current line 405.

24) Line 353: Abbreviation TWD missing.

Response: We have added the abbreviation of TWD at current line 383

25) Table 2: Why was a mean value used in TWD? Shouldn't there be season specific values?

Response: The reason we use a mean value to construct image composites for the TWD was to ensure that the analysis is not biased by short-term fluctuations or extreme seasonal variations, providing a stable representation of wetland conditions over time. Furthermore, there is presence of high cloud cover which limits the availability of season-specific images, making a mean composite the most practical alternative.

Round 1 Reviewer 1 attachement:

Comment

Thank you for considering me as a reviewer for this study on the large-scale fragmentation of wetlands across the African continent. The work presents significant content on the theme of wetland protection. Overall, this study provides important results, with well-written paragraphs and a logical flow. However, in my humble opinion, the new information provided still requires significant changes and additions. This raises questions about its suitability for publication in a journal like *Nature Communications*. Therefore, I propose **major revisions** to the article, with the possibility of submission to another Nature journal, such as Scientific Reports, for example.

Abstract:

Line 11-12: I am not sure that high-resolution map of Africa is lacking (Zhang et al., 2024, Yan and Niu, 2009; Anzhen Li et al., 2022, etc. have studied that question).

Line 15: Could you please add one or two sentences on the methodological approach used in the paper?

Line 20: modify the sentence please

Line 21: how did you find and write the value of C carbon for European wetlands? (12-31 Gt)

Line 39: more other activities are mentioned in African wetlands as highlighted in some papers, could you please revise this please

Line 57: missing information on why studying **wetland fragmentation**? Wetland fragmentation is a significant threat to the health and functioning of wetland ecosystems and highlights the importance of conservation efforts aimed at preserving and restoring these valuable habitats. Wetlands become divided or separated into smaller, isolated patches or fragments. This fragmentation can occur due to various human activities and natural processes, such as urbanization, agriculture, infrastructure development, and changes in hydrology. Did you consider that? OR highlight that in the introduction please

You speak about small wetlands, many studies have focused on small wetlands that you did not mention while could more be focused on. (Chula et al., 2023; Sakana et al., 2013; Mwita et al., 2012, etc.)

Line 60-63 : To be revised

Line 82 : Nothing in the south? South Africa, Zambia, Zimbabwe, Botswana, Tanzania, Tchad, Morocco, etc. What are you calling wetland complexes?

Line 58 : please specify the number of ground control for each wetland type and distribution

Specify, the source of the data map of climatic zones, the process used to estimate the carbon stock,

Line 139-140: please specify and be clearer please. All the values from 156 to 177 have to be justified and sourced.

Discussion:

There still more things to add in this discussion please.

Material and method

Still confusion on how you certified the value used. You may have chosen a date (2015), for example, because before this date, some older databases may include wetland areas that are no longer considered as such (many marshes, for example, have disappeared). Other wetland areas are so small and insignificant that they are not represented. Additionally, in some conflict-affected regions where it is difficult to obtain field data, there are limitations in these areas.

Line 246: Please specify how you find 16 spectral bands of 10m resolution for Sentinel 2. Specify also which type of Sentinel you used: 2A or 2B, etc.

Line 265: Here, I process.....

Globally the method is not clear, how you integrated and maintained the 10m resolution, how you combined the population and sentinel images with different resolutions.

Overall, this approach could not identified the inland valleys, floodplains, ponds, etc. wetlands. The seasonal wetlands also can not be identified.

Since wetlands are defined based on topography, hydrology, vegetation cover, etc. Why did you select only vegetation cover? Using Optical data from sentinel? Why did not use the DTM, DEM images with topographic indices such as slope, curvature, TWI, TPI, etc.

Please justify also why choosing the RF model that others. Why did not you use other accuracy coefficients such as AUROC, TSS, DeLong Test, etc.

Other factors such as distance to villages, city, road, etc. should be used to better highlight the population activities that are drivers of change in wetlands.

Line 365 to 400: Not clear, need to be as clear as possible.

Some sentence still need grammar check.